# O-Forge: An LLM + Computer Algebra Framework for Asymptotic Analysis

## Abstract

Large language models have recently demonstrated advanced capabilities in solving IMO and Putnam problems; yet their role in research mathematics has remained fairly limited. The key difficulty is verification: suggested proofs may look plausible, but cannot be trusted without rigorous checking. We present a framework, called LLM+CAS, and an associated tool, O-Forge, that couples frontier LLMs with a computer algebra systems (CAS) in an In-Context Symbolic Feedback loop to produce proofs that are both creative and symbolically verified. Our focus is on asymptotic inequalities, a topic that often involves difficult proofs and appropriate decomposition of the domain into the "right" subdomains. Many mathematicians, including Terry Tao, have suggested that using AI tools to find the right decompositions can be very useful for research-level asymptotic analysis. In this paper, we show that our framework LLM+CAS turns out to be remarkably effective at proposing such decompositions via a combination of a frontier LLM and a CAS. More precisely, we use an LLM to suggest domain decomposition, and a CAS (such as Mathematica) that provides a verification of each piece axiomatically. Using this loop, we answer a question posed by Terry Tao: whether LLMs coupled with a verifier can be used to help prove intricate asymptotic inequalities. More broadly, we show how AI can move beyond contest math towards research-level tools for professional mathematicians.

## 1 Introduction

Several fields of mathematics and computer science, like Analysis and Partial Differential Equations Hörmander (1983); Evans (2010), Analytical Number Theory Iwaniec & Kowalski (2004), and Theoretical Computer Science Cormen et al. (2009), require proving $O(\cdot)$ estimates, otherwise known as asymptotic inequalities. An asymptotic inequality, denoted by $f \ll g$, or equivalently by $f = O(g)$, implies that there exists a positive constant $C > 0$ such that $f \leq Cg$. In this paper, we use the notation $\ll$, also known as Vinogradov notation.

It is widely known that proving asymptotic inequalities can be quite involved. For example, the well-known Riemann Hypothesis is an asymptotic inequality of the following form: it states that if $\pi(x)$ is the number of primes that lie between 1 and $x$, then

$$\pi(x) - \int_2^x \frac{dt}{\log t} \ll \sqrt{x} \log x.$$

As another example, consider the following series:

$$S(h, m) := \sum_{d=0}^{\infty} \frac{2d + 1}{2h^2 \left(1 + \frac{d(d+1)}{h^2}\right) \left(1 + \frac{d(d+1)}{h^2 m^2}\right)^2} \ll 1 + \log(m^2).$$

Proving such estimates as a key step in a proof is often the bread and butter of analytical number theorists.

On the face of it, proving this estimate seems extremely difficult, and the non-specialist may not even know where to start. However, the old maxim of divide-and-conquer holds true here: if one is able to find a decomposition of this series into smaller, more manageable components, then proving this estimate may be trivial for those components Tao (2024). Putting these estimates together allows one to prove the estimate for the whole series. Hence, the primary difficulty lies in finding such a "nice" decomposition; once it is found, the proofs become trivial. Sadly, even professional mathematicians may find it difficult to "guess" these decompositions Tao (2024).

Similarly, consider the following asymptotic inequality, which is as asymptotic version of the famous Arithmetic Mean- Geometric Mean inequality Wikipedia contributors (2025):

$$(x_1 x_2 \dots x_n)^{1/n} \ll \frac{x_1 + \dots + x_n}{n},$$

where each $x_i \geq 0$. Proving such estimates can be non-trivial for $n \geq 3$. However, such proofs become absolutely trivial if the correct decomposition of the domain $\mathcal{D}$ is found. In fact, given the correct decomposition of $\mathcal{D}$, the proofs become so simple that theorem provers, like SMT solvers and Mathematica's `Resolve` function, are able to complete these proofs using only first-order logic.

Fields medalist Terence Tao has stated that an AI-powered tool that can suggest such appropriate decompositions, and then also prove the desired asymptotic inequality in each of those desired subdomains, can be extremely useful for research mathematicians in several fields(Tao, 2025a; 2024). In this paper, we present a tool aimed at realizing that vision.

In other words, we present an AI-powered tool that can quickly prove tricky estimates that may take research mathematicians several hours, thereby providing them with a useful research companion that can save them lots of time and effort.

## O-FORGE: AN LLM+CAS TOOL FOR ESTABLISHING ASYMPTOTIC INEQUALITIES

We build an end-to-end system that we call O-Forge, that takes as input a conjectured asymptotic inequality in latex format, and produces two outputs: (i) a decomposition of the domain into appropriate sub-domains, and (ii) whether Mathematica's `Resolve` function was able to prove this estimate in each subdomains.

(** describe the structure of the prompt**) In more detail, our system works as follows: (i) accepts a latex input from the user and prompts a frontier LLM to propose an effective decomposition of the domain into subdomains, and (ii) uses a computer algebra system (CAS) to produce rigorous proofs and hence certify the inequality over the whole domain. This proof is produced by the `Resolve` function in Mathematica via quantifier elimination in first-order logic. The primary benefit of using the `Resolve` function is that it is able to reliably prove estimates involving non-linear functions like log and exp, that SMT Solvers like Z3, CVC5 and MetiTarski are unable to.

At this point in time, the tool is not designed to produce a formal proof (say, in Lean 4). Having said that, Although the `Resolve` function only returns a True value if it is able to rigorously prove the estimate, there is an element of trust involved, as it does not produce a proof object that can be independently verified.

### 1.1 CONTRIBUTIONS.

- **O-Forge** An AI-powered tool that accepts as input a latex formula of an estimate, and outputs whether the tool has been able to complete a proof of the estimate (cf. Fig. 1). We have a dedicated website for this tool: o-forge.com.

  Frontier LLMs often provide incorrect proofs of these estimates, and manually spotting these errors can take a lot of time and effort. Our tool does away with this, and returns a "True" value only when the estimate has been rigorously verified. This can save mathematicians a lot of time and effort.

  Note that being able to simply visit a website, put in a formula in latex, and get as an output the proof status of the estimate, will be very helpful for mathematicians who are not comfortable with cloning GitHub repositories and running code from the command line.

- **Case study 1: Asymptotic inequalities**: Using our tool, we rigorously verify an asymptotic inequality proposed by Terry Tao.

  Such inequalities can often be non-trivial to prove, and standard tricks like Cauchy-Schwarz, Jensen's inequality, etc might not directly apply.

  We follow a novel algorithm, as proposed by Tao, of first splitting the domain into the correct subdomains, and then proving the estimate in each of those subdomains. The manner of this splitting is suggested by a frontier LLM. If the correct splitting is found, then the difficulty of the proof instantaneously changes from seemingly impossible to almost trivial, and the `Resolve` function is able to complete such proofs.

- **Case study 2: Series decomposition**: Using O-Forge, we rigorously verify a series estimate proposed by Terry Tao.

  We first prompt a frontier LLM to find the correct way of decomposing the series into manageable components, and then prove the estimate for each component separately. Put together, this gives us a proof of the estimate for the entire series.

  Again, we emphasize that proving such estimates without decomposing the series first would prove exceedingly difficult, and almost no theorem prover, human or machine, would be able to complete the proof.

Our primary novelty is in being able to automate proof completion for difficult research problems that should take most research mathematicians lots of time and effort. No existing AI tools are able to complete and symbolically verify proofs of this kind. Moreover, although frontier LLMs may be able to produce some of these proofs, these proofs are often incorrect, and need to be manually verified. Our tool does away with the need for manual verification.

## 2 FRAMEWORK: LLM-PROPOSED DECOMPOSITION + CAS VERIFICATION

We now elaborate on the steps mentioned in Fig. 1.

**Step 1: Latex input** The user inputs the conjectured estimate in the form of a Latex formula. This is especially useful for mathematicians who may not have prior experience with programming, and hence may be unable to run code from the command line.

**Step 2: Decomposition proposal.** The LLM proposes a finite cover $D = \bigcup_{i=1}^{k} D_i$ (or $0 = d_0 < d_1 < \cdots < d_k < \infty$ for series) guided by cues such as dominant terms and monotonic regimes. The proposal aims to localize each subproblem to simple comparisons (e.g., leading term domination, dyadic thresholds).

**Step 3: Regime-wise simplification.** For expressions with rational structure, we extract numerator/denominator leading behavior on each $D_i$, enforcing positivity where required to avoid spurious bounds. When denominators are not sums of positive terms, we guard against singular regions by refining the split.

**Step 4: Symbolic verification via `Resolve`.** For each $D_i$, we attempt

$$\forall x \in D_i \ : \ f(x) \le C \, g(x),$$

or in the series case, $S_{d_i, d_{i+1}} \le C \, g$, searching $C$ over a finite grid (e.g., 1 to $10^4$). Verification succeeds if `Resolve` returns `True`; the global inequality holds if all pieces return a "Proved" or "True" value.

This $C = 10^4$ value can of course be changed to an arbitrarily large number by the user. We keep it at this value because most of the proofs that mathematicians need in their research are completed for $C < 10$ (all the examples that we tested were completed for $C \le 2$).

---

[0]Mathematica's `Resolve` can often decide formulas involving log and exp using quantifier elimination over the reals, but it does not emit an externally verifiable proof object. SMT solvers like Z3/cvc5 have limited capabilities for proving lemmas involving transcendental functions.

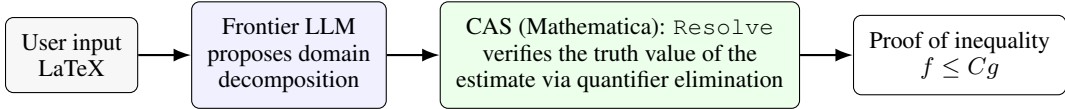

Figure 1: Workflow: The user puts in a conjectured estimate in latex notation; a frontier LLM proposes a domain decomposition; a CAS (`Resolve`) performs quantifier elimination to rigorously verify the required inequalities.

## 3 AN EXPLORATION OF ASYMPTOTIC ANALYSIS WITH O-FORGE

Mathematicians and computer scientists spend lots of time and effort proving asymptotic estimates in their day-to-day work. We present two case studies below Tao (2024).

CASE STUDY 1: ASYMPTOTIC INEQUALITY

Let us study the following estimate:

$$xy \ll x \log x + e^y, \tag{1}$$

where $x, y \in \mathbb{R}$, and $x \geq 1, y \geq 0$. This is equivalent to the fact that there exists a positive constant $C$ such that

$$xy \leq C(x \log x + e^y),$$

where the constant of course is independent of $x$ and $y$.

It is possible, with some analysis, to "guess" that such an estimate must be true. If $y$ goes to $\infty$ at roughly a comparable pace as $x$, then $e^y$ must dominate $xy$. If $y$ stays constant and $x \to \infty$, then $x \log x$ must eventually dominate $xy$. One may think up other possibilities, like both $x$ and $y$ going to $\infty$ but $y$ approaching $\infty$ much slower than $x$, and convince themselves that such an estimate must be true.

However, how does one go about proving this? One cannot prove this by using the standard tricks of proving inequalities like completing squares, Cauchy-Schwarz, etc. The standard method of proving this inequality is to find the correct way of decomposing the domain into sub-domains, and then proving this estimate in each of those sub-domains Tao (2024).

But what sub-domains should we choose? After some trial and error, one may finally find the following decomposition: $y \leq 2 \log x$ and $y > 2 \log x$. Within these sub-domains, proving this estimate becomes trivial. We demonstrate an extremely short proof:

$$y \leq 2 \log x \implies y \ll \log x \implies xy \ll x \log x \leq x \log x + e^y. \quad \square$$

$$y > 2 \log x \implies x \log x + e^y \geq e^{y/2} e^{y/2} \geq xy. \quad \square$$

As can be seen from the above proofs, the only "creative" step was to find the correct decomposition of the domain into the sub-domains $y \leq 2 \log x$ and $y > 2 \log x$. This decomposition is certainly not obvious at first, unless one has spent some time playing around with these inequalities. However, once one is able to correctly identify this decomposition, the math proofs in themselves are trivial, and can certainly be completed by a powerful computer algebra system without a human in the loop. We delegate the task of guessing the correct decompositions to frontier LLMs like Gemini and ChatGPT, which do a commendable job.

This approach is in part inspired from AlphaGeometry Trinh et al. (2024), where a specially trained LLM is asked to predict the next useful construction in the process of writing an IMO geometry proof, and then the proof is completed by an intricate proof system designed by the authors. It is our belief that currently, LLMs are much better at guessing "next steps" and heuristics than generating full proofs at one shot.

However, our work diverges from AlphaGeometry in two key aspects:

- There is no longer a need to train LLMs from scratch. This is informed by further work in Alpha Proof, where the Google DeepMind teams simply took an off the shelf LLM, and were able to perform reinforcement learning on it until it became good enough to win IMO gold. Frontier LLMs are already extremely powerful at Math, and we can leverage this prowess to 'guess" the correct decomposition of the domain.

- We use Mathematica's `Resolve` function to verify the truth value of the estimate in each subdomain. This function returns a True value only if it is able to complete a proof using quantifier elimination over the reals. Hence, one can be assured that if the Resolve function returns a True value, then the proof has indeed been completed.

  One major drawback of using LLMs for mathematical research is that the proofs that they present are often incorrect, and verifying them is very time intensive. Coupling a powerful LLM with a verifier like Mathematica does away with this painful verification process. If Mathematica returns "Proved", then the mathematician may be assured that the estimate is indeed true.

CASE STUDY 2: SERIES DECOMPOSITION

Let us now analyze the following series proposed by Terry Tao (2024):

$$S(h,m) := \sum_{d=0}^{\infty} \frac{2d+1}{2h^2 \left(1 + \frac{d(d+1)}{h^2}\right) \left(1 + \frac{d(d+1)}{h^2 m^2}\right)^2} \ll 1 + \log(m^2). \qquad (2)$$

Such estimates come up regularly in analytic number theory. However, proving it directly may seem almost impossible.

The way that one generally attacks such problems is by breaking this series up into different components, such that proving this estimate for each component may be trivial. For instance, a rigorous training in analysis may inform the reader that the natural breaking points for this series are $\{\lceil h \rceil, \lceil hm \rceil\}$. This is for the following reasons:

- if $d$ lies between 0 and $\lceil h \rceil$, then the summand can be approximated as $\frac{d+1}{h^2}$.

- if $d$ lies between $\lceil h \rceil$ and $\lceil hm \rceil$, then the summand can be approximated as $\frac{1}{d}$.

- If $d$ lies between $\lceil hm \rceil$ and $\infty$, then the summand can be approximated as $\frac{h^4 m^4}{d^5}$.

In all the above cases, the sum of such approximations over their respective ranges can be trivially shown to be $\ll 1 + \log m^2$.

Hence, there are two non-trivial creative jumps. The first is to "guess" the correct decomposition of the series into sub-series, and then the second step is to find the correct simplification of the summand in each regime, so as to be able to prove the estimate.

We use a frontier LLM to "guess" the correct decomposition, and use elaborate Mathematica code to find the correct simplification of the summand in each regime. We don't use LLMs for the latter purpose for the following reasons:

- Frontier LLMs are not always reliable at finding these simplifications. Making API calls to Gemini, for example, only sporadically gave us the correct simplifications.

- Finding these simplifications is equivalent first simplifying the summand, and then finding the leading order term in both the numerator and the denominator. Clearly, if the numerator and denominator are a sum of finite numbers of terms, then the summand $\ll$ ratio of these leading order terms. Because we use the `Resolve` function to choose the leading order terms, we can be guaranteed that we're getting the correct answer.

- The final proof of the estimate is completed by Mathematica anyway.

In a well-defined sense, the accuracy of the LLM output is the bottleneck; and robust design principles indicate that one must minimize the number of bottlenecks in any workflow. Therefore, we only

prompt the LLM once in the entire process, and the rest of the proof completion is carried out by Mathematica. In fact, we now have a domain `o-forge.com`, where you can try and prove these inequalities and series estimates yourself.

### CHOICE OF COMPUTER ALGEBRA SYSTEM

We choose the `Resolve` function in Mathematica for the following reasons:

One may use Lean, and powerful Lean tactics like "aesop" and "simp" to complete proofs of trivial statements. In fact, there exist powerful technologies like Lin et al. (2025) that can suggest useful tactics in order to semi-automate proof-generation. However, we found such tactics woefully inadequate for proof completion when transcendental functions like $\log$ and $\exp$ are involved. For example, even in Terry's attempt to create such a tool Tao (2025b), he is using the linarith tactic in Lean, which cannot complete proofs for non linear functions.

The `Resolve` function, on the other hand, is able to complete such proofs with ease.

One may also use SMT solvers for proof completion; however, there are severe limitations in using those. Z3, which is the most popular SMT solver currently, is unable to handle transcendental functions. CVC5 and MetiTarski, which are able to handle transcendental functions, were not able to reliably complete even the simplest proofs. For example, both CVC5 and MetiTarski were unable to complete the following proof:

$$\log x \leq \log y \implies \exp(x) \leq \exp(y).$$

Taking inspiration from AlphaGeometry, we also tried to write down functions that could carry out algebraic manipulations in order to complete proofs. However, the space of algebraic manipulations is much larger than in IMO geometry. Hence, creating such a list would almost certainly be prohibitive.

We then learned about Mathematica's `Resolve` function, and it was surprisingly able to prove almost every simple estimate we threw at it. Moreover, because `Resolve` uses quantifier elimination over the reals, it returns True only when it is able to "legally" complete a proof.

One drawback of using a closed source software like Mathematica is that it does not produce a proof object that can be externally verified. However, based on our experiences of testing O-Forge with several estimates, we believe that Mathematica's `Resolve` function is the superior option for this particular application.

Note that Maple also has a powerful proof completion tool called `QuantifierElimination`. However, it has the same drawback as Mathematica, in that it doesn't produce a proof term. SageMath also has something similar called `qepcad(...)`. However, it is nowhere as powerful as `Resolve`.

## 4 IMPLEMENTATION

We first make an API call to a frontier LLM to find the correct decomposition. We use a structured prompt so as to get the correct answer reliably:

```
<code_editing_rules>
    <guiding_principles>
    -
    </guiding_principles>

    <task>
    -
    </task>

    <requirements_for_breakpoints>
    -
    </requirements_for_breakpoints>

    <output_format>
```

```
        _
      </output_format>
    </code_editing_rules>
```

We then pass the output from the LLM into Mathematica. The API call to a locally run copy of Mathematica simply consists of various modules and commands. We provide a short snippet below.

```
      Resolve[ForAll[{series.other_variables},
      _
      logForm["Resolve results", res2];

      If[AllTrue[res2,TrueQ],True,res2]
```

The result from the Mathematica computation consists of True/False statements for each subdomain of the decomposition. If "True" is returned for each subdomain, the code prints "Proof verified".

The package also includes a simple CLI:
`decomp prove question_<id>` for inequalities
`decomp series series_<id>` for series decompositions.

Internally, `llm_client.py` proposes splits; `mathematica_export.py` invokes `wolframscript` to run `Resolve`. A lightweight overview with examples is provided in Anonymous (2025).

## 5 EMPIRICAL EVALUATION

In addition to the above-mentioned case study of hard problems, we tested our tools on an extensive suite of around 40-50 easier problems, in order to study how well it performs on a diverse set of inequalities.

This dataset consisted of problems like proving the estimate that

$$\sum_{n=1}^{\infty} \frac{1}{n^p} \ll 1$$

if $p > 1$, proving the estimate that $\sum_{n=1}^{\infty} r^n \ll 1$ if $|r| < 1$, etc.

We observe the following:

- Generally, for a 2 or 3 variable function, a small number of decompositions ($k \leq 4$) is sufficient for proof completion by the CAS. This is surprising, as there may be problems which require a very large number of decompositions. However, our experience was that the number of decompositions grows linearly with the number of variables, although there is no a priori reason to expect that.
- Subdivisions based on orderings of the variables are common, and mostly robust, especially for functions that are symmetric in all variables;
- Regime-wise leading-term replacement is sufficient for the computer algebra system to be able to complete proofs. Without this simplification, Mathematica's `Resolve` function falters. For example, without this simplification, Mathematica tries to find a closed form expression for the series in terms of `gamma` functions, and is then unable to complete proofs using `Resolve`.

In summary we observe that our approach is robust, and is able to prove a wide variety of asymptotic inequalities.

## 6 RELATED WORK

**LLM+CAS framework.** AlphaGeometry Trinh et al. (2024) is a pioneering tool that uses LLMs to come up with the "creative" step in mathematical problem solving, and then using a symbolic

verifier to actually complete the proof. The one drawback in AlphaGeometry is that it is prohibitively difficult to generalize it to domains that are not as constrained in scope as IMO plane geometry. For instance, generalizing AlphaGeometry to prove asymptotic estimates would be very challenging, as the space of possible algebraic manipulations is much larger than that of plane geometry calculations.

*Key differences*: We bypass that problem by using frontier LLMs and versatile computer algebra systems like Mathematica, which, put together, are able to convincingly solve a wide variety of problems right out of the box.

**Lean tactics.** Terence Tao has also contributed to a tool Tao (2025b) via which they are able to generate Lean proofs for linear estimates; they do so by using several powerful Lean tactics like "Linarith". Such approaches have the advantage of being able to generate proofs certificates. On the other hand, they are unable to deal with functions like $\log$ or $\exp$.

*Key differences*: We extend this work greatly by being able to prove estimates for a much more general class of functions like transcendental functions, and also being able to verify estimates for series like Eq. (2).

**Autoformalization.** LLMs have also been fundamentally useful in autoformalization via tools like GoedelProver and Kimina-Autoformalizer Lin et al. (2025); AI-MO (Project Numina) (2025). Autoformalization can certainly be a powerful tool in the near future for proving the estimates that are currently being proven by the `Resolve` function. However, the current autoformalization technologies have primarily focused on contest math, and are unable to reliably generate proofs for research-level math; we find Mathematica's `Resolve` function to be better suited to such tasks.

**Positioning our contributions.** O-Forge makes several novel contributions relative to other AI for Math tools:

This is one of the first AI-powered tools that is useful for research-level mathematics today. A lot of development in the mathematical capabilities of LLMs has focused on making them better at contest mathematics like the IMO or Putnam. However, those capabilities have not extended to research-level math. O-Forge is able to prove estimates that research mathematicians spend considerable time and effort proving on a regular basis.

Using frontier LLMs for research purposes can be frustrating, as they often provide incorrect proofs with a high degree of confidence, and such proofs need to be painfully checked before they are recognized to be incorrect. Our LLM+CAS framework avoids this problem by directly verifying the estimate on every subdomain suggested by the LLM. Hence, a human-in-the-loop is not needed.

The setup itself is completely painless, and suitable for mathematicians who may lack coding and other computer skills. Being able to just go on to `o-forge.com`, put in a latex formula, and check whether their conjectured estimate is correct, will hopefully lead to a rapid adoption by the mathematical community.

# 7 LIMITATIONS AND FUTURE WORK

**Proof objects.** `Resolve` does not produce proof objects that can be independently verified, but only carries out symbolic verification. The `Resolve` function only returns a "True" value if has been able to complete a proof using quantifier elimination over the reals. However, we acknowledge that there is still an element of trust involved; that a closed-source company like Wolfram is indeed performing the correct manipulations "under the hood".

For our purposes, we are unable to use other technologies because none of them are able to complete non-trivial proofs like `Resolve` is. However, we hope that autoformalization becomes powerful enough in the future that we are able to delegate such proof completions to such a tool.

**Summand upper bounds.** Currently, we simplify the summand of the series by extracting the leading order term from both the numerator and the denominator. This may not be valid simplification for more complex summands, and perhaps performing RLHF on an off-the-shelf LLM would allow us to obtain correct simplifications more generally.

## 8 REPRODUCIBILITY

Code and CLI usage, along with worked examples, are available at Anonymous (2025). The repository requires Python 3.9+, access to a locally run copy of Mathematica via `wolframscript`, and access to a frontier LLM via an API key (details in the README).

We also have a user-friendly website: `o-forge.com`. This accepts inputs in latex format, and outputs the truth value of the estimate after producing a rigorous proof using quantifier elimination over the reals.

## 9 ETHICS STATEMENT

We gladly acknowledge the ICLR Code of Ethics.

This work requires access to expensive technologies like Mathematica and frontier LLMs. The costs involved may be prohibitive for researchers outside of university systems with access to such tools.

## 10 CONCLUSION

Mathematical arguments can often be reduced to verifying routine but time–consuming estimates. For analysts and theoretical computer scientists, these often take the form of asymptotic inequalities, and proving these can take several hours and days.

The difficulty here is two-fold: first, the correct decomposition of the domain or series must be found, which in most cases can be highly non-trivial. Second, the estimate must be verified in each subdomain, which requires lots of simplification and heuristic arguments.

We present O-FORGE, which prompts a frontier LLM to propose domain decompositions, and then validates the asymptotic estimate symbolically with Mathematica's `Resolve` function. This provides mathematicians with a useful tool that can do the tedious job of verifying these research-level estimates for them. In moving beyond contest math, we present a tool that can be genuinely useful for mathematical research.

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

## A    USE OF LLMS IN PAPER PREPARATION

We have used LLMs to collect .bib references of papers, help make a TikZ diagram, and to discuss the structure of the paper. We have also used it to troubleshoot some issues in our code, as well as polish some portions of the writing.

## B    APPENDIX: MINIMAL USAGE

```
pip install -r requirements.txt
export WOLFRAMSCRIPT=/path/to/wolframscript # if needed
```
Add a Gemini API key in the .env file Add a problem to examples.py, then run decomp prove question_X or decomp series series_Y.

You may also use the website o-forge.net

