# OpenReview forum: "O-Forge: An LLM + Computer Algebra Framework for Asymptotic Analysis"
_ICLR.cc/2026/Conference — Submitted to ICLR 2026_

### Official Review · Reviewer_x7mF · 2025-10-14

**Soundness:** 1
**Presentation:** 1
**Contribution:** 1
**Rating:** 0
**Confidence:** 5

**Summary:**

The paper introduces **O-Forge**, a workflow that couples a frontier LLM with a computer algebra system (CAS), specifically Mathematica’s Resolve, to prove asymptotic inequalities by (i) asking the LLM to propose a domain (or index) decomposition and (ii) using Resolve to verify each piece via first-order quantifier elimination. A toy inequality and a series bound from Tao are used as case studies; the paper claims the approach “moves beyond contest math” by offloading the creative split to the LLM and the verification to CAS. The system exposes a CLI and a website front-end; evaluation is described as ~40–50 “easier problems,” with qualitative takeaways about typical numbers of subdomains and the usefulness of leading-term simplifications.

**Strengths:**

N.A.

**Weaknesses:**

There are too many drawbacks in this paper. In general, this submission is more like a blog post instead of a rigorous paper as it lacks of enough and solid experiments and evaluations to demonstrate its claims. Some of the weaknesses are as follows.
- **Lack of Novelty:** LLM + CAS could be viewed as part of Tool-use LLM research. The proposed idea is not novel.
- **Insufficient rigor and experimental substance:** The “evaluation” consists of two main case studies plus ~“40–50 easier problems,” but there are no well-defined benchmarks, success metrics, or failure analyses (rates of correct/incorrect splits, wall-clock, query counts, sensitivity to prompts, comparisons to baselines like hand-crafted heuristics or SMT-based pipelines, etc.). As is, this reads more like a prototype report/blogpost than an ICLR-level empirical study.
- **Underspecified method details:** Key pieces are missing or skeletal. For example, the “structured prompt” is left with placeholders (“describe the structure of the prompt”), and the code snippet for Resolve is fragmentary; the search over constants C is a coarse grid without justification or sensitivity analysis. This makes the approach hard to reproduce or evaluate scientifically.
- **Heavy reliance on closed-source Mathematica without proof objects:** While Resolve is powerful, the paper acknowledges there is no proof term and asks the reader to trust a closed system; this undermines the claim of “rigorous verification,” especially for research-level math. No attempt is made to cross-check with open tools (e.g. SageMath) on a subset, or to export certificates.
- **Reproducibility & accessibility concerns:** Running the system requires Mathematica and a frontier LLM API, making reproduction costly; even the authors’ ethics section notes access costs. There is a website, but that can’t substitute for open artifacts or independent verification.
- **Scope creep vs. precise problem definition:** The paper oscillates between “asymptotic inequalities” and the specific case study. I could feel the motivation of this paper: making a true AI4Math tool or project to better help professional mathematicians instead of some fuzzy LLMs that could only do math questions. But there is no crisp task definition, no sanity check for the motivation, and no quantitative experiments to show the helpfulness and effectiveness. This vagueness makes it hard to judge whether this is something scientifically helpful or just some course project.

**Questions:**

- Please use \citep for non-subject/object citations.

---

> ### Author Response · Authors · 2025-11-26
> **Response to reviewer x7mF**
>
> Reviewer x7mF says that the project lacks novelty, is not appropriately benchmarked, lacks details on the prompt design, that we don't test other alternatives like SageMath, that we don't include artifacts, and that we don't define the problem appropriately.
>
> We feel that this review is based on a very cursory reading of the paper, without any understanding of the difficulty of the math problems involved. Note that Terry Tao and his collaborators have tried solving this problem too (check out his GitHub). However, only our approach is powerful enough to solve research-level inequalities. Our main innovation is a comprehensive simplification of the input function using multiple pages of Mathematica code, before feeding the simplified function into Resolve[]. We arrived upon this workflow after testing several SMT solvers and related tools. We only include a skeleton of the prompt because everything else is a placeholder for the details of the input function. We have tested SageMath, and have in fact written about it in the paper. Resolve[] is much more powerful than the alternatives. We do in fact include all artifacts with the paper, and it is truly surprising that the reviewer could not access clearly attached documents (we also explicitly state in the paper that we are attaching all artifacts). We don't quite understand the last point, because we clearly state in the introduction that we are going to prove asymptotic inequalities, and we go on to do that in the rest of the paper; not sure how we oscillate between such inequalities and the case study.

---

> > ### Comment · Reviewer_x7mF · 2025-11-26
> >
> > I would say this might be one of the worst AI4Math works I have ever reviewed. The authors are not familiar with what a standard ICLR paper is and do not provide quantitative evidence to support their proposed methods and rebuttals. This is not about "understanding the difficulty of the math problems." (Tbh, I said in my review that I sort of felt the motivation of it, but due to the writing and incompleteness of this work, it's hard to appreciate it.) The authors should conduct sufficient analyses and clearly present them in the paper, rather than asking reviewers to directly examine their artifacts or code. For example, authors should present statistical evidence (tables, graphs) to demonstrate the effectiveness of their method (success rates, etc.), and then delve into specific cases, illustrating how success is achieved. Additionally, the authors should not just use "we did/experimented/explored/tested" as evidence, which I believe should be the consensus for an academic publication.
> >
> > Lastly, there are some useful AI review tools [1] that can identify these weaknesses in our reviewers' reviews, even before the authors submit. In general, I would say the motivation and vision behind this paper are great, as it could really help mathematicians in their research workflow, which I call a true AI4Math work. However, I wish the authors could make it more substantial, as the current version feels like a blog post.
> >
> > [1] https://review.cspaper.org/

---

### Official Review · Reviewer_3RYN · 2025-10-31

**Soundness:** 1
**Presentation:** 1
**Contribution:** 2
**Rating:** 0
**Confidence:** 4

**Summary:**

Following Terry Tao’s proposed algorithm, a system for asymptotic analysis is implemented. It combines an LLM for the creative part of sub-domain choice, and uses Mathematica for the rest of the steps - notably “Resolve” for proofs in specific sub-domains.

2 non-trivial asymptotic bounds were proven via this system.

**Strengths:**

If works well, applicable to a broad range of scientific endeavors.

Final results are grounded via a reliable CAS system.

Shown proof of concept for useful mathematical results.

**Weaknesses:**

Line 86: “(** describe the structure of the prompt**)”...
This is unprofessional at best.

Line 91: All the mentioned solvers need citations, as well as the Mathematica Resolve function - due to its central role in your system.

Line 100: Fig.1 is mentioned initially on page 2 but appears on page 4. Why?

Line 101 (and multiple others): “o-forge.com”.
At the top right corner of the front page of this website it clearly states:

Created by

Vijay Ganesh

Ayush Khaitan

Violating the ICLR guidelines regarding anonymity.


Regarding the website itself, while I liked the UI and readily available examples, it sometimes returns weird results.
For example, Example Series 1 returns a summary that contains a single word: “The”. I ran it a few times to make sure it’s not some random LLM aberration.
Other times the output shows a Python error stack trace (“Execution Failed”).
Seems underbaked and the code is unstable.

Line 109,113,119 (and others): Citations. You’re mentioning Prof. Tao many, many times - but he is a prolific mathematician. Point to the specific works you use.

Line 123: While I personally agree with this claim on (most) interesting series bounds, it needs to be quantified (benchmark vs. other methods), or at least supported by multiple examples from the literature.

Line 127: The novelty claim is unclear to me. It seems like an engineering project - implementing a (good) idea by Prof. Tao. While I agree that such an online tool can be useful for mathematicians around the world - and would like to encourage you to improve it and fix the bugs - the system does not constitute *scientific* innovation done by you.

The generate (via LLM) -> verify (via CAS) approach was done in multiple projects (though not specifically in your use case) - so the core approach is also not novel on its own.

If the strongest results are the proof of the specific use-case (which to my understanding is indeed novel, but I don’t know enough to say how impactful this singular result is), then perhaps consider submitting to a mathematical / experimental math journal?

Line 295: I’m a supporter of readable papers and not-too-formal language, but this is too much.
The paper is not your blogpost.

Line 348: What do you mean “around 40-50”??

Section 5: You need concrete statistics for all the claims here. Currently it’s anecdotal.

Section 6: There are multiple other works in AI for Math that apply combinations of CAS/code generation + LLMs. The current overview is quite limited.

**Questions:**

Line 256: This “elaborate Mathematica code’ seems like an important part of the system - perhaps ~50% of its power?

Line 270: Once in the entire process? If I’m reading the outputs on your website correctly, there are recursive attempts to re-define the sub-domain partition, until you get “True” on all of them?

Line 328: How is the output passed to Mathematica? Is there a specific format that you force it to use in its output? Do you parse the output text and translate it to Mathematica code? If so, how?

Line 359: How do you count decompositions? Number of separate domains or number of boundaries?

Line 367: Any ideas why this happens?
How do you do this replacement?

**Details Of Ethics Concerns:**

The project website “o-forge.com” is mentioned multiple times in the paper.
At the top right corner of the front page of this website it clearly states:

Created by

Vijay Ganesh

Ayush Khaitan

Violating the ICLR guidelines regarding anonymity.
It also shows a link to the paper on arXiv, which contains author affiliations.

---

> ### Author Response · Authors · 2025-11-26
> **Response to reviewer 3RYN**
>
> Reviewer 3RYN states that the presentation is not always careful, that the tool website reveals the names of the co-authors, that the particular example of Series 1 sometimes returns a strange answer, that there is little evidence of scientific innovation in the project, that the language is not always formal, and that the section referring to other work in the field is not comprehensive.
>
> We thank the reviewer for their review, and their very helpful advice. We agree that the presentation can certainly be improved in several parts, and we plan on following the reviewer's advice regarding this. We also plan on removing our names from the website. Note that the output for Series 1 is correct (if you read the bottom panel, it says inequality proved). It's just that the LLM is unable to summarize the long output, with details on each subdomain, and that is why the strange output is seen (we hope to correct this as well).
> Note that this problems were pursued by Terry Tao and his collaborators (check out his GitHub), but ours is the only approach that is powerful enough to prove these kinds of research level questions. The main scientific innovation here is the several pages of Mathematica code that we wrote, customized for asymptotic inequalities, that takes in a complicated algebraic expression and systematically simplifies it to only one or two terms, before those terms are fed into the Resolve function. The Resolve function is unable to prove things about complicated functions; it is our custom simplification procedure, in addition to the Resolve function, that makes the toolchain work. We arrived upon this workflow after experimenting with several SMT solvers (including variants like MetiTarski that can deal with transcendental functions) and other tools like SageMath for months.

---

> > ### Comment · Reviewer_3RYN · 2025-11-26
> >
> > I do not doubt that there was substantial work invested here and see the potential for value. But, as the work stands at the moment, even if you rewrite the paper, remove the names, fix the code etc - it will still not fit a top conference like ICLR, for 2 main reasons:
> >
> >  - As you say yourself, "The main scientific innovation here is the several pages of Mathematica code that we wrote". This is where the main power of the system lies. This is not AI, in the sense that the method was maticulaously hand crafted. Perhaps consider a publication in algebraic tools? Or follow in the footsteps of SageMath or the SMT solvers you've tried? Or - if the mathematical results themselves are important enough - try a mathematical journal (depends on whether there is an interesting proof that can be generalized from your current results)? In general it's hard to distill the *scientific* (vs. engineering) innovation that your work proposes.
> >
> >  - Many (all?) or your claims are either anecdotal or demonstrated on a limited number of examples. You say that you tried experimenting with other workflows and tools? Great - show it. Describe the exact experiments and compare results. Can the few examples you've solved be seen as a part of a family of math problems? Can you run the different methods on e.g. 500 of them, and show results? Try reading your paper from the standpoint of a skeptic - you must defend each claim.
> >
> > I wish you the best of luck in your next submissions.

---

> > > ### Author Response · Authors · 2025-11-26
> > >
> > > Thanks, that's a fair assessment. Just to respond to some specific claims:
> > >
> > > 1. The solution is not really hand-crafted for the particular examples in mind. When I say Mathematica code, I do not mean that I have written an explicit solver that works only for those examples. Extracting the leading order term of an expression is a standard technique that works for all inequalities. Ultimately, the behavior of an inequality is determined by the leading order term (at least asymptotically, which is the focus of this paper). It is this that we do with the Mathematica code. Hence, this technique is fairly general. We demonstrate this on the website by including several varied questions.
> > >
> > > 2. This is fair. Just to elaborate here on failed results, although we clearly didn't do a good job of explaining this in the paper: a) SMT solvers are not great at proving even basic inequalities. For example, they're not able to prove that if a < b, then exp(a)< exp(b). Even tools like MetiTarski are terrible at this. b) SageMath has some useful tools, but the Resolve[] function worked much better for our use case.
> > >
> > > We plan on taking all your advice, and implementing it in a future (hopefully much more polished) version of the paper. As you can probably guess, this version was certainly rushed. However, we do thank you for taking the time to offer some amazing advice, and giving us lots to chew on.

---

### Official Review · Reviewer_5WVK · 2025-11-01

**Soundness:** 2
**Presentation:** 2
**Contribution:** 2
**Rating:** 2
**Confidence:** 4

**Summary:**

he paper presents O-Forge, a system that integrates a large language model (LLM) with a computer algebra system (CAS) to aid in the verification of asymptotic inequalities. This is an instance of classical synthesis paradigms such as oracle guided inductive synthesis combining an inductive LLM with deductive reasoning system to generate formal artifacts. The paper describes this as an “In-Context Symbolic Feedback Loop”.  The LLM proposes domain decompositions (i.e., how to split a problem into manageable subdomains). The CAS (via Mathematica’s Resolve function) then verifies whether each subdomain satisfies the proposed inequality using first-order logic and quantifier elimination.

**Strengths:**

The authors claim their tool can handle research-level asymptotic inequalities, going beyond standard competition-style problem solving by combining LLM creativity with CAS rigor. This is left to some subjective interpretation.
Two case studies: an asymptotic AM-GM inequality and a series decomposition, are used to demonstrate the concept.

**Weaknesses:**

The paper reads more like a concept demo or blog post than a rigorous scientific study, lacking detailed quantitative and ablation studies with proper baselines. Testing on self-curated  "suite of around 40-50 easier problems" and a few case studies falls far short of the evaluation expected of a research paper.

Hybrid symbolic–neural approaches (e.g., Lean+LLM, AlphaProof, Autoformalization pipelines; see https://arxiv.org/abs/2310.17807, https://ieeexplore.ieee.org/document/10356332) have already explored the same broader paradigm with planning, code generation, formal proof verification, not just heuristic checking. The authors present O-Forge as “revolutionary,” yet it is essentially prompting an LLM for suggestions and sending them to a formal tool - Mathematica.

Crucial implementation details are omitted. How exactly is the LLM prompted? How is the “in-context feedback” loop structured? How is the decomposition quality evaluated or improved iteratively? Are there failure modes where the LLM produces incorrect decompositions, and how are these handled?

**Questions:**

Can you expand experimental evaluation and share quantitative metrics (success rate, runtime, size of inequalities handled) over larger benchmark suite to substantiate performance?

---

> ### Author Response · Authors · 2025-11-26
> **Response to reviewer 5WVK**
>
> Reviewer 5WVK says that the article lacks ablation studies, and that the tool is tested on a self-curated set of 40-50 problems. They also state that several such tools already exist, and that this tool lacks novelty, and that the paper does not mention how the LLM is prompted, how in-context feedback is provided, the failure modes of the LLM, etc.
>
> We would like to thank reviewer 5WVK for providing helpful feedback. However, we would like to push back against the claim that several such tools already exist: none of the existing tools (including the ones mentioned by the reviewer) can solve research level problems in analysis. And we say this from experience; we have, in the process of solving these problems, tried most of the existing tools. This tool is designed very specifically, keeping analysts in mind, and it required a lot of custom planning and tooling in order to be able to solve such intricate problems. Moreover, this tool does not simply take the output from an LLM and feed it into Mathematica. There are several intermediate stages involved, in which our custom Mathematica code simplifies the function until it is reduced to its leading order term. It is this leading order term that is then fed to the Resolve[] function. All of this is discussed in the paper. We also do provide the exact prompt that we feed the LLM on page 7.

---

> > ### Comment · Reviewer_5WVK · 2025-11-27
> > **Unclear novel contribution**
> >
> > The rebuttal appears to reiterate what is already in the paper and has failed to address the primary concerns of the reviewer. The most optimistic read of the paper would suggest that there are some cool engineering tricks but the reviewer went over the paper again, and did not get a clear idea of such an engineering/empirical value that practitioners could easily adopt. Perhaps, a rewrite of the paper would help. In its current form, the reviewer cannot recommend the acceptance of this paper.

---

### Official Review · Reviewer_y2Zz · 2025-11-03

**Soundness:** 1
**Presentation:** 1
**Contribution:** 1
**Rating:** 0
**Confidence:** 4

**Summary:**

The paper introduces a CAS + LLM system for proving asymptotic expressions. The primary thrust of the system is to use a LLM to propose decompositions and then use a CAS to test their correctness.

**Strengths:**

Due to the very unusual nature of the weaknesses, I don't have any strengths to comment on.

**Weaknesses:**

This paper does not appear to contrain any evidence as to its effectiveness. Section 5 begins "In addition to the above-mentioned case study of hard problems, we tested our tools on an extensive suite of around 40-50 easier problems, in order to study how well it performs on a diverse set of inequalities," but there is no mention of these "hard problems" anywhere in the paper. Additionally, for these easy problems, the paper presents three high level takeaways but no actual evidence.

**Questions:**

Did I misunderstand something? Does this paper present evidence of its correctness?

---

> ### Author Response · Authors · 2025-11-26
> **Response to reviewer y2Zz**
>
> Reviewer y2Zz says that due to the very unusual nature of the weaknesses, they don't have any strengths to comment on. They claim that the paper does not contain any evidence of its effectiveness, and claim that there is no evidence of "these hard problems" anywhere in the paper. They also state that the paper has three takeaways, but no evidence.
>
> We are saddened by this review. Almost the entirety of the paper focuses on discussing how our tool O-Forge solves very specific and extremely difficult problems in asymptotic analysis. Terry Tao talks about how difficult these problems are on his blog. Note that these are questions asked by research mathematicians on MathOverflow, and that such questions generally refer to research-level questions in the asker's own field. We isolate three problems and discuss them across multiple sections, and fully solve them using our setup. In fact, one of those problems is given on the first page itself! Due to the unusually terse nature of this review, we are having a hard time thinking of a helpful response. We would love to hear back from the reviewer, as we do want to improve our tool to perhaps fit their use case.

---

> > ### Comment · Reviewer_y2Zz · 2025-11-26
> >
> > In your paper you explicitly claim to have run evaluation on many easier problems, but don't share the actual results of those evaluations. You don't event report the success rate of your method, let alone compare that to other methods.
> >
> > Originally I thought that the three hard problems were illustrative examples of the set of hard problems with the full results withheld, much like the results on the easier problems are. Rereading the paper in light of your reply and exchange with Reviewer 3RYN, I'm unsure if you ran the methodology on more than the three hard problems specified in the paper.
> >
> > When reviewing a new methodology in machine learning (whether as a peer reviewer or just as a reader of a paper), important questions to consider are:
> > 1. How much do I trust the way they evaluated their methodology to reflect the performance in use-cases I care about?
> > 2. How successful according to their metrics are they, in absolute terms.
> > 3. How successful according to their metrics are they, relative to the previous best approaches.
> >
> > If your main evidence is success on three examples, that makes me worried about #1. I'm not saying you engaged in any wrongdoing, but it's certainly plausible that you could have tried many examples and only reported on the three that made you look best. We usually expect people to report on a large, broadly sampled set of problems. For example you could randomly sample 1,000 problems on math overflow (maybe filtering out some with tags like "big list") and report what percentage of those problems you're able to solve. The results will likely be worse than 100%, but if your method gets 20% and others get 2% that's still evidence of a huge advance and would be much more compelling than
> >
> > Additionally, withholding details of results while eluding to them is a huge red flag. If you say you ran on a set of X problems the bare minimum you must do is report the success rate of your method and comparable ones. I would not allow any paper that contains a sentence like "we ran on 40-50 easier problems" and doesn't report these things to pass peer review. It is also weird that you don't state the exact number. That's something you obviously know, why not share it?

---

> ### Author Response · Authors · 2025-11-26
>
> Thank you for your response.
>
> "Originally I thought that the three hard problems were illustrative examples of the set of hard problems with the full results withheld, much like the results on the easier problems are. Rereading the paper in light of your reply and exchange with Reviewer 3RYN, I'm unsure if you ran the methodology on more than the three hard problems specified in the paper." This response betrays that the we have done a less than ideal job of communicating our work.
>
> The problems that we worked on are contained in the artifacts attached with the paper, as well as our website, which we mention several times in the paper.
>
> The website that we explicitly mention several times contains several varied problems, and the GitHub repo that we have attached contains more. The examples are representatives of several problems that we try, and not the only problems that we try on. In particular, we have included every single problem that Terry Tao had talked in his public posts on this this topic.
>
> We agree that we should have benchmarked our results better, but note that there do not exist any benchmarks in this area. Finding thousands of research-level problems in asymptotic analysis is surely a more difficult task than finding thousands of olympiad level problems or undergraduate level problems, but we plan on correcting this in the near future. Note: there are probably less than 100 open problems in asymptotic analysis, which is the real benchmark for a tool of this kind. The two problems that we focus on (and we have tested our tool on around 50 more, which are included on our website and artifacts) are open problems in asymptotic analysis on MathOverflow.

---

### Author Response · Authors · 2025-12-04

We would like to offer a summary of the discussion on this paper. We feel that the reviewers are fair in their assessment of the paper as lacking in some fundamental benchmarking that makes for a scientific document, although they do commend the mathematical content of the paper. However, benchmarking is intrinsically difficult for a tool that deals with research mathematics, since most existing benchmarks concern competition problems or undergraduate-level material. With that in mind, we still find the reviewers largely fair and accurate in their assessment, and we hope to implement several of their helpful suggestions in our future work, including a future version of this paper.

**Lack of benchmarking**

We feel that this was the biggest issue raised by the reviewers, and they have brought it up in several ways. First, they mention that we state we tested the tool on 40 to 50 problems, but do not mention what these problems are. These problems are included in the artifacts as well as on the website referenced in the paper, although we could have highlighted this more clearly. They also note that we have not benchmarked our tool against known datasets. This is mainly because such benchmarks do not currently exist for research-level inequality problems. Our focus in this submission was on the problems suggested by Terence Tao, all of which we were able to solve. In the near future, we hope to build a benchmark ourselves to help fill this gap.

**Lack of novelty**

Some reviewers commented that our tool lacks novelty, interpreting it as simply taking the output of an LLM and putting it into Mathematica. This does not fully capture what our system does. Our custom code takes a function as input and simplifies it in multiple stages until we are left with only leading order terms. It is these terms that are then put into Mathematica, along with the domain decompositions suggested by the LLM. Notably, our method is able to prove several sophisticated inequalities that appear to be beyond the reach of existing computational tools. We will do more in the future to describe this part of the pipeline clearly.

**Violating anonymity**

One reviewer noted that we had not fully respected anonymity because our names appeared on the tool’s website. We have taken note of this and removed identifying information.

**Not included artifacts**

Some reviewers stated that we did not include documentation or artifacts. We did include the GitHub repository, dataset, and other materials, but we understand that this may not have been as visible as intended. We will work to make this clearer.

Although we are not in complete agreement with all the reviewers’ points, we believe they have raised several valid criticisms of our paper and have taken the time to express them carefully. We value their time and effort in reviewing our work, and we will do our best to implement their helpful suggestions in the next version of the paper.

---

### Meta-Review · Area_Chair_HoqB · 2026-01-04

**Summary:**

The paper proposes a framework called "O-Forge" that integrates Large Language Models (LLMs) with Computer Algebra Systems (CAS), specifically Mathematica. The core objective is to automate the proof of asymptotic inequalities, a task relevant to research-level mathematics. The system utilizes an "In-Context Symbolic Feedback loop" where the LLM proposes domain decompositions (splitting the problem into sub-domains), and the CAS verifies the inequalities within those domains using quantifier elimination. The authors highlight the application of this tool to specific problems posed by mathematician Terence Tao.

**Reviewer Concerns:**

**1. Severe Lack of Empirical Evidence and Benchmarking**
The most critical deficiency identified by all reviewers is the absence of rigorous evaluation.

* The paper mentions testing the tool on "40-50 easier problems" but fails to present any data, success rates, or analysis regarding these tests within the paper itself.


* Reviewers noted that withholding results while alluding to them is a significant methodological red flag.


* The evaluation relies almost entirely on anecdotal case studies (three hard problems) without broader quantitative metrics. As Reviewer y2Zz noted, the paper does not appear to contain evidence of its effectiveness.



**2. Presentation and Readiness**
The submission appears to be in a draft state, lacking the polish required for a top-tier conference.

* **Placeholders:** The text contains unprofessional placeholders such as "(** describe the structure of the prompt**)", indicating the paper was not finished prior to submission.


* **Documentation:** Key implementation details, such as the specific prompting strategies and the "elaborate Mathematica code" central to the contribution, were underspecified.


* **Tone:** Multiple reviewers remarked that the paper reads more like a "blog post" than a scientific study.



**3. Anonymity Violation**
The submission violated ICLR's double-blind reviewing policy. The paper repeatedly references a project website ("o-forge.com") which explicitly listed the authors' names and affiliations during the review period. The authors acknowledged this oversight in their response.

**4. Novelty and Contribution**
Reviewers questioned the scientific novelty of the work.

* The paradigm of  (neurosymbolic approaches) is already established in the literature (e.g., AlphaProof, Lean+LLM).

**Reviewer Scores:**

0 2 0 0

---

### Decision · Program_Chairs · 2026-01-26

Reject